# Eight Typical Aroma Compounds of ‘Panguxiang’ Pear during Development and Storage Identified via Metabolomic Profiling

**DOI:** 10.3390/life13071504

**Published:** 2023-07-04

**Authors:** Huiyun Li, Chaowang Ma, Shunfu Li, Huimin Wang, Lisha Fang, Jian Feng, Yanmei Wang, Zhi Li, Qifei Cai, Xiaodong Geng, Zhen Liu

**Affiliations:** 1College of Forest, Henan Agricultural University, Zhengzhou 450002, China; huiyun8303@126.com (H.L.); nuannuanxx0210@163.com (S.L.); whm126126@126.com (H.W.); lishafang0210@163.com (L.F.); 15713679647@163.com (J.F.); wym3554710@163.com (Y.W.); lizhi876@163.com (Z.L.); cai_qifei@henau.edu.cn (Q.C.); 2Zhengzhou Zheng Shi Chemical Co., Ltd., Zhengzhou 450002, China; chaowang618@163.com

**Keywords:** pear, flavor, metabolome, aroma

## Abstract

Aroma is an appreciated fruit property, and volatile flavor plays a key role in determining the perception and acceptability of fruit products by consumers. However, metabolite composition that contributes to the aroma in fruit quality is unclear. In this study, we detected 645 volatile organic compounds of ‘Panguxiang’ pear in total, including esters, alcohols, alkanes, acids, ketones, terpenes and aldehydes. In addition, the levels of sugars, organic acids and amino acids in ‘Panguxiang’ pear were investigated using high-performance liquid chromatography. In the aroma generation, glucose was the dominant sugar, followed by sucrose and fructose. At the development transferred storage stage, organic acids may not participate in aroma biosynthesis. The amino acids that may play potential roles in aroma substance synthesis are tyrosine and glycine. Through metabolomics analysis at different stages of ‘Panguxiang’ pear, we selected 65 key metabolites that were significantly related to glucose, sucrose, fructose, tyrosine and glycine, according to the trends of metabolite concentrations. Finally, we chose eight candidate metabolites (e.g., three esters, two aldehydes, one alcohol, one acid and one ketone) as the representative aroma substances of the ‘Panguxiang’ pear compared to the metabolome of the ‘Korla’ at stage Z5. Data and results from this study can help better understand the variations in aroma quality among pear varieties and assist in developing breeding programs for pear varieties.

## 1. Introduction

Pear (*Pyrus* spp.) is one of the most important fruit crops with high economic value in temperate zones and is cultivated in more than 50 countries [1]. As one of the most important fruit crops in China, it grows in moderate climate zones with many cultivars and is believed to have originated in the western mountainous areas [2]. Pear is widely distributed and cultivated in north and south China, with the largest production and richest germplasm resources in the world. The ‘Panguxiang’ pear, a new maturing variety bred from the Biyang Piao pear, has a plentiful and unique aroma. It is popular among consumers for its unique fragrance, subtle aroma, sweetness and crispness.

As aroma is one of the most appreciated fruit properties, volatile flavor compounds may play a key role in determining the perception and acceptability of products by consumers [3,4]. Aroma, an important trait of fruit quality, has received increasing attention in recent years. To date, the aromas of various fresh fruits have been evaluated, including white pear [5], apple [3], watermelon [6], mango [7], melon [8], strawberry [9], peach [10] and tomato [11]. Aroma comprises more than eight classes, including amino acid-derived compounds, phenolic derivatives, esters, terpenoids, alcohols, aldehydes, ketones and alkenes, of which esters are the most important volatiles of pear fruits [4,12,13]. The identification of key volatile flavor metabolites that carry the unique traits of natural fruits is essential, as it provides the principal sensory identity and characteristic flavor of the fruits [14]. Soluble sugars and organic acids strongly affect pear flavor, and volatile organic compounds (VOCs) determine pear aroma [15]. Moreover, some of the aroma metabolites are derived from sugars as the substrate [16,17]. In recent years, rapid advances in high-throughput broad-targeting metabolomics have made it possible to characterize the chemistry of flavor [15]. However, the connection between sugars or amino acids and aroma metabolites has not been considered.

To better identify the typical metabolites of pear during development and storage, here we identified the volatile compounds and detected the concentrations of soluble sugars and amino acids at the nine stages of the ‘Panguxiang’ pear. The aroma characteristic metabolites of pear were evaluated by integrating the metabolomes, soluble sugars and amino acids. Then, through comparison to the metabolome of ‘Korle’ pear at 28 days after storage, we identified eight typical metabolites of the ‘Panguxiang’ pear aroma. In this work, the production of aroma volatile compounds and concentrations of certain related organic matters were assessed throughout the fruit development and storage of the ‘Panguxiang’ pear. The general aim was to study the ability to produce aroma volatiles that are important for improving fruit aroma quality in this pear cultivar.

## 2. Materials and Methods

### 2.1. Plant Materials and Experimental Design

The experiment was carried out in the Key Laboratory of Forest Resources Cultivation of the State Forestry Administration in Zhengzhou, Henan Province, China. The tested soil was sandy loam soil at pH 7.0. Six-year-old ‘Korle’ and ‘Panguxiang’ pear trees with strong growth were chosen as the experimental trees in 2021. The 30th day after flowering (DAF) was marked as S0. Uniformly sized fruits were picked from the middle or upper periphery of each tree. In addition, all fruits were bagged with opaque paper bags that were yellow outside and black inside. Then, samples were collected every 30 days, beginning at S0 (S1, 60DAF; S2, 90DAF; S3, 120DAF; S4, 147DAF). After ripening and harvesting, some pears were stored in a plant growth chamber, which was maintained at 20 °C and 70% humidity. Samples were taken on the first day of storage (1DOS, Z1) and then collected every seven days subsequently (Z2, 7DOS; Z3, 14DOS; Z4, 21DOS; Z5, 28DOS). The outer pericarp with the skin of each fruit was cut into small pieces, rapidly frozen in liquid nitrogen and then stored at −80 °C for further metabolic and transcriptomic analysis. All experiments were analyzed with three biological repeats, and each repeat consisted of 15 fruits of uniform size.

### 2.2. Metabolite Extraction and LC-MS

Pulp and skin were mixed and collected from all stages for metabolomic analysis. Three biological replicates were used, and 1 g (1 mL) per stage of the powder was transferred immediately to a 20 mL headspace vial (Agilent, Palo Alto, CA, USA), containing a NaCl saturated solution, to inhibit any enzyme reaction. The vials were sealed using crimp-top caps with TFE-silicone headspace septa (Agilent). At the time of SPME, each vial was placed at 60 °C for 5 min, and then a 120 µm DVB/CWR/PDMS fiber (Agilent) was exposed to the headspace of the sample for 15 min at 100 °C.

After sampling, the VOCs from the fiber coating were desorbed in the injection port of the GC apparatus (Model 8890; Agilent) at 250 °C for 5 min in the splitless mode. The VOCs were identified and quantified using an Agilent Model 8890 GC and a 7000D mass spectrometer (Agilent) equipped with a 30 m × 0.25 mm × 0.25 μm DB-5MS capillary column (5MS: 5% phenyl-polymethylsiloxane). Helium was used as the carrier gas at a linear velocity of 1.2 mL·min^−1^. The injector and the detector were kept at 250 and 280 °C, respectively. The oven temperature was programmed from 40 °C (3.5 min), increasing at 10 °C·min^−1^ to 100 °C, at 7 °C·min^−1^ to 180 °C, at 25 °C·min^−1^ to 280 °C and holding for 5 min. Mass spectra were recorded in the electron impact ionization mode at 70 eV. The quadrupole mass detector, ion source and transfer line temperatures were set at 150, 230 and 280 °C, respectively. MS in the ion monitoring [11] mode was used to identify and quantify the analytes.

### 2.3. Bioinformatics Analyses of Metabolome and Transcriptome

Principal component analysis (PCA), a statistical procedure that models the variation in terms of its principal components, was performed using statistics function prcomp within R. Differentially expressed metabolites (DEMs) between groups were determined at variable importance in projection (VIP) ≥ 1 and absolute log2FC (fold change) ≥ 1. VIPs were extracted from the results of orthogonal partial least-squares discrimination analysis (OPLS-DA), including score plots and permutation plots, which were generated using R package MetaboAnalystR. The data were processed via log-transformation (log2) and mean centering before OPLS-DA. A heatmap with dendrograms was produced using Pheatmap. The correlation coefficients between soluble sugars, organic acids or amino acids and metabolites were calculated on Hmisc. The metabolites were subjected to K-means clustering with an Mfuzz package with version 2.56.0. A Kyoto Encyclopaedia of Genes and Genomes (KEGG) pathway enrichment of genes was performed using the hypergeometric distribution-based R program (version 3.6.0.). The networks were visualized via a Cytoscape, version 3.9.1 [18].

### 2.4. Soluble Sugar, Organic Acid and Amino Acid Contents in Pear

The same batch of samples for metabolomics and transcriptomics was taken to determine physiological indicators. Soluble sugars and organic acids were extracted from 2 g of pulp, along with the amino acid. The organic acids of fruits were determined via liquid chromatography, and metaphosphoric acid was used as the leaching solution for organic acids. The soluble sugars and organic acids were detected with an Ultimate 3000 UV detector (Thermo Fisher, Waltham, MA, USA). The contents of 17 amino acids were measured using an automatic amino acid analyzer. All data were generated from three biological replicates.

### 2.5. Statistical Analysis

Results were displayed as mean ± standard error (SE). SPSS 24.0 (SPSS, Chicago, IL, USA) and Microsoft Excel 2021 was used to implement a one-way analysis of variance (ANOVA) on three biological replicates. All plots were optimized for layout on Adobe Illustrator 2021.

## 3. Results

### 3.1. Changes of Metabolome in Pear during Development and Storage

Significant differences in the metabolites of pears among different stages were detected. PCA is a common method to assess the omics context and reduce the dimensionality of large data sets. The results of PCA show that the fit degrees of different biological replicates in the same stage are good, indicating that the samples in this experiment are representative. Except for the similarity between the fruit development of Z1 and S stages, the fruit development was quite different between the Z2-5, S1-4 and Z1 stages. In the first principal component (PC1) with a variation interpretation of 74.63%, all samples were divided into two parts, indicating that in terms of metabolic substances, the contents of aroma substances in the fruits during the growth period and the later storage period were quite different. However, in the second principal component (PC2) with a variation interpretation of 15.39%, there was no significant difference among fruits at different stages (Figure 1A). Notably, the sum of the variation explanation of PC1 and PC2 reached 90.02%, indicating that the differences in all samples distributed in PC1 were the most reasonable. Therefore, the PCA results show that the contents of pear metabolites in each stage are stable and can provide a good theoretical support for the search of key metabolites.

The number of DEMs in adjacent stages is an effective way to digitally measure the difference between two stages (Figure 1B). The total number of DEMs in all comparison groups was 172. In the comparison group of adjacent stages, the difference in development stages is stable, while the difference in storage stage fluctuates largely. The trend in the number of up-regulated differential metabolites is consistent with that of the total differential metabolites. Specifically, changes in the S2-Z1 process were relatively stable. The numbers of differential metabolites in the Z1_vs_Z2 and Z3_vs_Z4 groups were the largest and both increased significantly. The number of down-regulated differential metabolites decreased gradually in the development stage and remained stable in the storage stage from S1-Z5. In the S1_vs_S2 comparison group, 90% of the metabolites were down-regulated. In the Z1_vs_Z2 comparison group, the total number of differential metabolites and the number of up-regulated differential metabolites were both at the peak, but the number of down-regulated metabolites accounted for only 17.07%. These results indicate that fruit metabolism is active in the early stage of pear development. The metabolism of pear fruits changed the most at the early stage of storage, and then the changes tended to be stable.

The main purpose of metabolomic analysis is to detect and screen out biologically and statistically significant metabolites from biological samples and thereby to clarify the metabolic process and change mechanism of organisms. According to the results of PCA and correlation analysis, S1, S2, S3, S4 and Z1 can be combined into group R, and Z2, Z3, Z4 and Z5 can be combined into group S (Figure 1A). Then, the two groups were analyzed independently. An Upset map can uniformly show unique or common differential metabolites between different groups (Figure 1C,D). In group R (Figure 1C), the most unique differential metabolites in the comparison group of the adjacent stages were found in S2_vs_S3 (57), and the least number of differential metabolites was detected in S4_vs_Z1 (46). The S2_vs_S3 group and the S1_vs_S2 group contained at most 16 and only 7 specific differential metabolites, respectively. Notably, the number of differential metabolites shared by the four comparison groups was only four. In the group of adjacent stages, the most unique differential metabolites were S1_vs_S2 and S2_vs_S3, indicating that the metabolic process of pear fruits in S1 stage was relatively intense during development from stage S2 to stage S3. At the same time, the number of unique differential metabolites was the least in S3_vs_S4 and S4_vs_Z1, indicating that the metabolic changes of pear fruits were relatively slow when they were close to maturity and at the beginning of storage. In group S (Figure 1D), Z3_vs_Z4 had the largest stage specific difference, followed by Z4_vs_Z5 and Z2_vs_Z3. However, there were only two differential metabolites among the three comparison groups in group S, suggesting that the metabolic changes of pear fruits at different storage stages were different in terms of time. In the comparison group of the adjacent stages, the most unique differential metabolite was Z2_vs_Z3, indicating that the substance metabolism of pear fruits fluctuated greatly in the middle stage of storage.

### 3.2. Classification and Trend of Metabolites in Pear during Development and Storage

The 434 identified metabolites were integrated and classified (Table 1), including 97 esters, 91 terpenoids and 66 hybrid compounds. The hydrocarbon compounds encompass aromatic hydrocarbons and alkanes, including 34 alkanes and 19 aromatic compounds. More accurate key metabolic genes for further mining can be found in previous research results. The pear aroma substances in the compounds were screened for rearrangement and divided into five categories, including esters (97), terpenoids (91), hydrocarbons (both alkane and aromatic hydrocarbons, 53 in total), paraffin oxides (ketones, alcohols, aldehydes and acids, 109 in total) and others. The subsequent analysis mainly focused on the first four categories of metabolites.

The pear metabolic processes in the development and storage stages showed significant differences. To increase the accuracy of candidate metabolites and reduce false positives, the metabolites with similar concentration trends were screened (Figure 2). To improve the accuracy of metabolite screening, the esters were also divided by Mfuzz clustering into four categories. Of these, the numbers of metabolites in cluster 1 and cluster 4 were the largest (both 34). The same Mfuzz clustering was used to divide the hydrocarbon metabolites into four groups, and the number of metabolites contained in each group was almost equal: 13 (cluster 1), 16 (cluster 2), 14 (cluster 3) and 10 (cluster 4).

A total of 97 ester compounds were identified in the metabolome. The ester compounds were divided by the content change pattern into four clusters after heat map clustering analysis (Figure 2B). Each cluster has a specific temporal signature. As for the time variation pattern of cluster 2, the content of S1 was the lowest, which increased continuously to about stage Z1 and then decreased continuously. Unlike cluster 2, the contents of metabolites in cluster 4 increased almost continuously from stage S1 to Z5. However, the trend of metabolite contents in cluster 3 was completely opposite to those in cluster 4, showing a continuous downtrend from stage S1 to Z5. In terms of the number of clusters, the contents of ester metabolites during the development and storage of pear were mainly divided into two types. In type 1, the growth and development period increased, while the storage period decreased (cluster 2 containing 46 metabolites). In type 2, the contents of metabolites in S1-Z5 increased continuously and peaked at the end of storage (cluster 4 containing 36 metabolites).

A total of 53 metabolites were obtained by combining alkanes and aromatic hydrocarbons. According to different time trends, the hydrocarbon metabolites were divided into four classes using heat map clustering. From S1 to Z5, the number of metabolites in cluster 3 increased gradually (47.17%), which changed gently in the early development stage of the pears and dramatically during the storage period. The changing trends of only seven metabolites in cluster 1 were the opposite to cluster 3. These accounted for 35.85% of the hydrocarbon metabolites in cluster 4, and the metabolism of pear was gentle at the beginning of development and the end of storage but changed dramatically at the end of ripening and the beginning of storage. Cluster 2 contained only two metabolites with high content in stage S2.

Terpenoids are characteristic substances in the aroma components of pear, and 91 terpenoid metabolites were detected in the metabolome of the ‘Panguxiang’ pear. The terpenoids were divided into four groups by heat map clustering (Figure 2D). Cluster 3 contained the most abundant terpenoids (47), with the lowest content at stage S1, which gradually increased with growth and development, peaked at the end of development or early storage, and finally decreased. Moreover, the content of cluster 4 (32) was less in the growth stage but gradually increased after storage. The last two groups did not have obvious characteristics in content changes.

Alkane oxides include ketones, alcohols, aldehydes and acid metabolites. These metabolites were divided into four groups via the clustering analysis of the stage changes of a heat map (Figure 2E), among which the most obvious trend was found in cluster 2 (39) and cluster 4 (52). The common point of these two trends was that the contents of metabolites in S1 were low level in the developmental stage and increased during development or storage. Cluster 4 showed a single peak at late development or early storage. Meanwhile, the metabolites in cluster 2 had similar data expression forms, which were low in the growth stage and increased with the extension of storage time. In contrast, metabolite contents in cluster 2 reached a high level in the middle or later storage, while metabolite contents in cluster 3 reached a peak in the end of development or early storage and then decreased. The metabolite contents in cluster 1 (12) were maximized at the S1 stage and then decreased gradually with the prolongation of development or storage time. In total, these data showed four main trends in the content changes of all metabolites, and we selected metabolites with rapid improvement in storage period as candidate aroma substances.

### 3.3. Screening and Determination of Key Aroma Metabolites

After the trend classification of the above four categories of metabolites combined with the aroma-appearing stage of the ‘Panguxiang’ pear, we found that the expressions of the selected metabolites were low in the growth stage but increased sharply in the middle and later stages of storage, and this trend was stable. Finally, 119 key metabolites were obtained (Appendix A), including 34 esters, 25 hydrocarbons, 28 alkane oxides and 32 terpenoids. In combination with the number of candidate compounds and the concentration trend of each sample, it can be inferred that the main components in the aroma are esters and terpenoids. According to the metabolites, the esters, hydrocarbons, alkane oxides and terpenoids accounted for 35.05%, 47.17%, 25.45% and 35.16%, respectively, of the total compounds after screening. In conclusion, especially in the hydrocarbon compounds, nearly half of the metabolites may be the aroma components in the ‘Panguxiang’ pear. Terpenoids and esters were the second most common compounds, and one third of their metabolites were probably primary components of aromatic substances. Finally, nearly a quarter of the alkane oxide oxides have the potential to converse pear aroma. Therefore, the screening of key metabolites can be more accurate through the functional differences of these 119 metabolites.

### 3.4. Identification of Key Aroma Generation Substrates

To further clarify the pivotal aroma metabolites among development and storage stages of pear, we measured the contents of four soluble sugars, four organic acids and seventeen amino acids (Figure 3A,B). The contents of fructose and glucose continuously increased, while the concentrations of sucrose constantly declined from stage S1 to Z5. However, the contents of sorbitol, citric acid, shikimic acid, malic acid and quinic acid were not affected by the pear’s situation (Figure 3A). In addition, only Met content rose progressively from stage S1 to Z5, whereas four amino acids, namely Tyr, Gly, Thr and Lys, were reduced gradually (Figure 3B). Remarkably, Phe was up-regulated in the development stages but down-regulated in the storage stages. These results indicate that three sugars (e.g., fructose, glucose and sucrose) and six amino acids (e.g., Met, Tyr, Gly, Thr, Lys and Phe) are the substrates for the biosynthesis of pear aroma substances.

Next, regarding the unique aroma metabolites in the ‘Panguxiang’ pear, we further measured the significant correlations between 4 sugars or 4 organic acids and 119 metabolites (Figure 3C). Results showed the changes in the concentrations of 113 metabolites were highly synchronized with these 9 aroma substrates. To identify the key substances in pear aroma, we found thirty hub nodes, including twelve esters, nine oxides, seven hydrocarbons and two terpenoids, in this network (Figure 3C; Appendix A). Moreover, the neighbors/degree number of glucose, sucrose and fructose were all more than 30, indicating these sugars play key roles in pear aroma biosynthesis. Only the degree of Tyr was more than 30, and the Gly was significantly correlated with 26 metabolites, implying that Tyr and Gly contributed to pear aroma production. In total, these findings reveal that glucose, sucrose, fructose, Tyr and Gly are the synthetic substrates in the pear aroma synthesis pathway; in total, 65 metabolites correlated with these three sugars and two amino acids, which are the key ingredients of the ‘Panguxiang’ aroma (Appendix A).

### 3.5. Eight Characteristic Aroma Substances of the ‘Panguxiang’ Compared to the ‘Korla’

To explore the characteristic aroma substances of the ‘Panguxiang’ compared to the ‘Korla’ pear, we further measured the concentrations of soluble sugars, soluble acids and amino acids of the ‘Korla’ pear at stages Z1 to Z5 and the metabolome of ‘Korla’ pear at stage Z5. As shown in Figure 4A, sorbitol content was dramatically reduced from stage Z3 to Z5, whereas other soluble sugars and acids were not affected by pear storage. Notably, the contents of nine amino acids in total, namely Glu, Ile, Ala, Phe, Lys, Met, Leu, His and Pro, gradually increased from stage Z1 to Z5 (Figure 4B). The concentrations of Thr and Asp strikingly decreased at stage Z2 and then rapidly increased at stage Z3. These data indicate that amino acids rather than soluble sugars and acids play key roles in the aroma biosynthesis of the ‘Korla’ pear.

To clarify the differences of aroma components between the ‘Panguxiang’ and the ‘Korla’, we measured the metabolites of the ‘Korla’ at stage Z5 (ZK) and found that the all samples were clearly clustered into two groups based on pear varieties in PCA (Figure 4C). At stage Z5, 100 differentially expressed metabolites (DEMs), including 76 up-regulated metabolites and 24 down-regulated metabolites, were found in the ‘Panguxiang’ compared to the ‘Korla’ (Figure 4D; Appendix A). To confirm the characteristic aroma substances of the ‘Panguxiang’, we compared the key metabolites in Appendix A and the up-regulated DEMs in the ‘Panguxiang’ compared to the ‘Korla’. Results showed that only eight metabolites were shared between these two groups, including three esters, two aldehydes, one alcohol, one acid and one ketone, indicating that only a few metabolites belong to the characteristic aroma substances (Figure 4E,F, Table 2). Remarkably, these metabolites, except for NMW0070, were not enriched in KEGG pathways, indicating that the aroma biosynthesis pathway was not clear. Hence, these eight metabolites are the typical aroma ingredients of the ‘Panguxiang’ pear.

## 4. Discussion

Fruit aroma is composed of a complex combination of numerous volatile compounds and represents a major quality attribute of pear fruits, accounting for 10^−4^ to 10^−7^ of fruit mass [19,20]. Volatile aromatic substances are formed dynamically in fruits, and their types and concentrations change over time, such as D145, XMW1069 and NMW0130, but most of the aromatic substances are formed in later stages, including KMW0580, XMW0088 and KMW0530 [13,21] (Figure 2). Moreover, the composition of aroma is specific to the species of pear [4]. Volatile aromatic substances in pears have been extensively studied. For example, in four additional cultivars (e.g., ‘Xiang-mian Li’, ‘Mu-tou Su’, ‘Dang-Shan Suli’ and ‘Nan-Guo Li’ pear), aldehydes and alkenes are formed during early stages but alcohols and esters gradually increase in abundance as time progresses and the total aroma substances accumulate [21]. Reportedly, 121 metabolites were identified and quantified, including 40 esters, 32 alcohols, 16 aldehydes, 13 alkenes, 11 ketones and 4 acids in 12 cultivars of pear fruits [13]. Moreover, 221 major volatile components were detected in 202 mature pear cultivars, among which aldehydes, esters and alcohols were the most dominant [20]. In the present study, we detected 434 metabolites from 14 types in ‘Panguxiang’ pears in total, and the majority of these metabolites consisted of esters and terpenoids (Table 1, Appendix A).

Sugars and aroma are essential quality traits and have induced a great deal of research into the physiological properties of fruit aromas [17]. In pear leaves, sorbitol as the main photosynthetic product is transported through the phloem and accumulates in the fruits. Here, during the growth and storage stages of pear fruits, the sorbitol content is always the highest of all sugars, followed by fructose and glucose in the ‘Panguxiang’ pear (Figure 3A). In comparison, fructose content is the highest, followed by sorbitol and glucose, in the ‘Korle’ pear (Figure 4A). Notably, the content of sucrose was lower and gradually declined in both the ‘Panguxiang’ pear and the ‘Korle’ pear. However, in the Chinese white pear, sorbitol accounts for the largest proportion of total sugars at the early stage but then declines. In mature pears, sucrose and sorbitol account for about 30% and 20% of total sugars, respectively, with a smaller amount of fructose [17]. Hence, among various sugars, fructose, glucose and sucrose primarily affect the flavor of the ‘Panguxiang’ pear.

Volatile compounds are mainly derived from the metabolism of fatty acids, amino acids and carbohydrates [4,17]. The availability of primary precursor substrates is crucial for the biosynthesis of these aroma compounds, including sugars, fatty acids and amino acids. In most fruits, the major precursors of aroma volatiles are either saturated or unsaturated fatty acids [17,22]. Fatty acid-derived straight-chain alcohols, aldehydes, ketones, acids, esters and lactones, ranging from C1 to C20, are important and characteristic aroma compounds that are responsible for the intense flavor of fresh fruits [17]. Within pears, sugars, as precursors, are used in the biosynthesis of many quality compounds and flavor components, such as vitamins and aromatic substances [17]. The main types of volatile aromatic substances formed through carbohydrate metabolism are furaneols, pyrones and various terpenes [17]. Moreover, esters, terpenes, and aldehydes were the main volatile metabolites in pears [20,23,24]. In this study, we predicted nine aroma synthesis substrates, including fructose, glucose, sucrose, Met, Tyr, Gly, Thr, Lys and Phe, and obtained a total of 65 volatile candidate metabolites (Figure 3; Appendix A). Then, eight volatile aroma metabolites were obtained as the characteristic aroma substances of the ‘Panguxiang’ pear through comparison with the metabolic composition of the ‘Korle’ pear (Figure 4E; Table 2). Thus, we suggest that the accumulation of volatile esters, aldehydes, ketones and acids in pears is important for improving fruit edibility.

## 5. Conclusions

Aroma is an important indicator to reflects the quality of a pear. In this paper, volatile organic compounds of the ‘Panguxiang’ pear were primarily esters, alcohols, alkanes, acids, ketones, terpenes and aldehydes. Although the number and classes of metabolites identified in pear fruits at different stages were the same, the concentration of volatile metabolites varied significantly between the developmental and storage stage. Moreover, glucose, sucrose and fructose were the dominant sugars in the ‘Panguxiang’ pear fruit, and tyrosine and glycine were the most important amino acids. Finally, from sixty-five metabolites significantly correlated with these three key sugars and two pivotal amino acids, we speculate that esters and aldehydes are the characteristic aroma substances of the ‘Panguxiang’ compared to the ‘Korla’ pear. The evaluation of volatiles at different stages in terms of their effects on fruit aroma is aimed at improvements in fruit quality and is useful for future breeding.

## Figures and Tables

**Figure 1 life-13-01504-f001:**
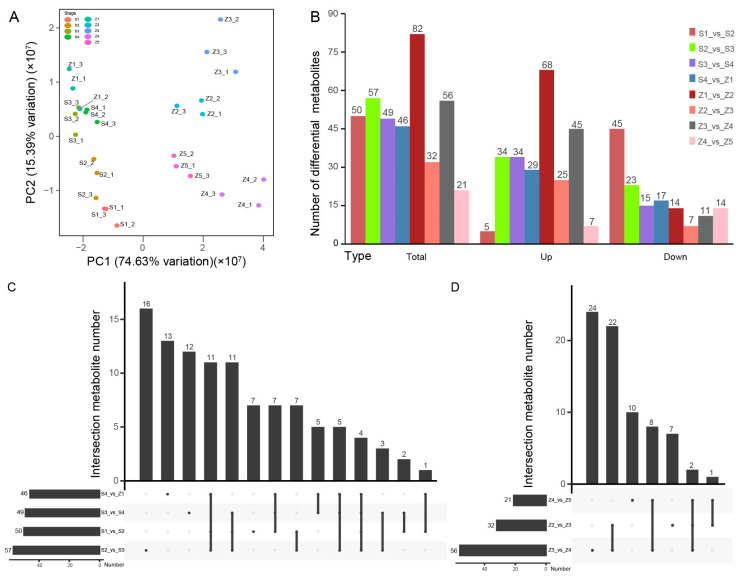
Differences in aroma components of pear fruits at different stages. (**A**) PCA of metabolomics. (**B**) Total distribution of differential expressed metabolites (DEMs) in different comparison groups. (**C**,**D**) Numbers of common and specific DEMs among different comparison groups.

**Figure 2 life-13-01504-f002:**
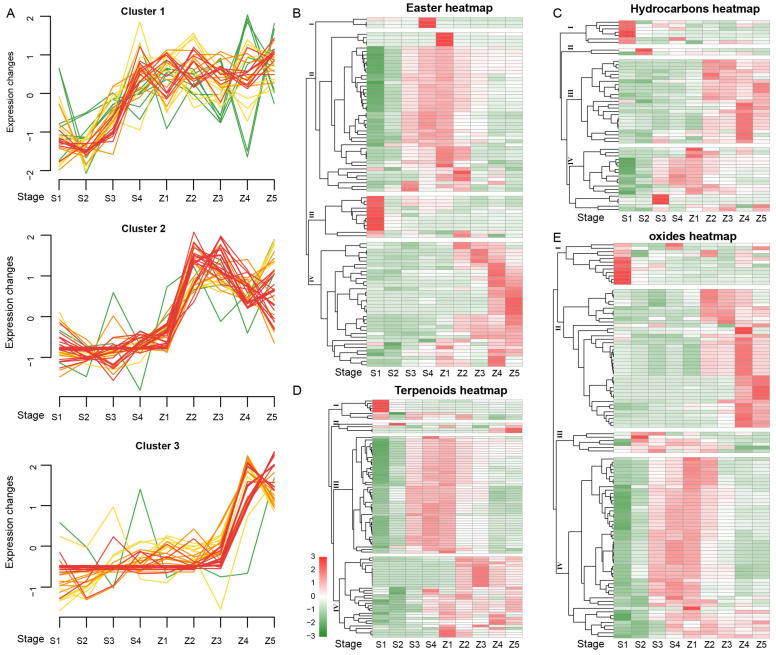
Grouping of the concentrations of aroma components in pear fruits at different stages. (**A**) K-means analysis of all metabolites. The green, yellow, and red color represent low, middle, and high membership value with mean value per cluster, respectively. Heatmap of metabolites belonging to esters (**B**), hydrocarbons (**C**), terpenoids (**D**) and oxides (**E**).

**Figure 3 life-13-01504-f003:**
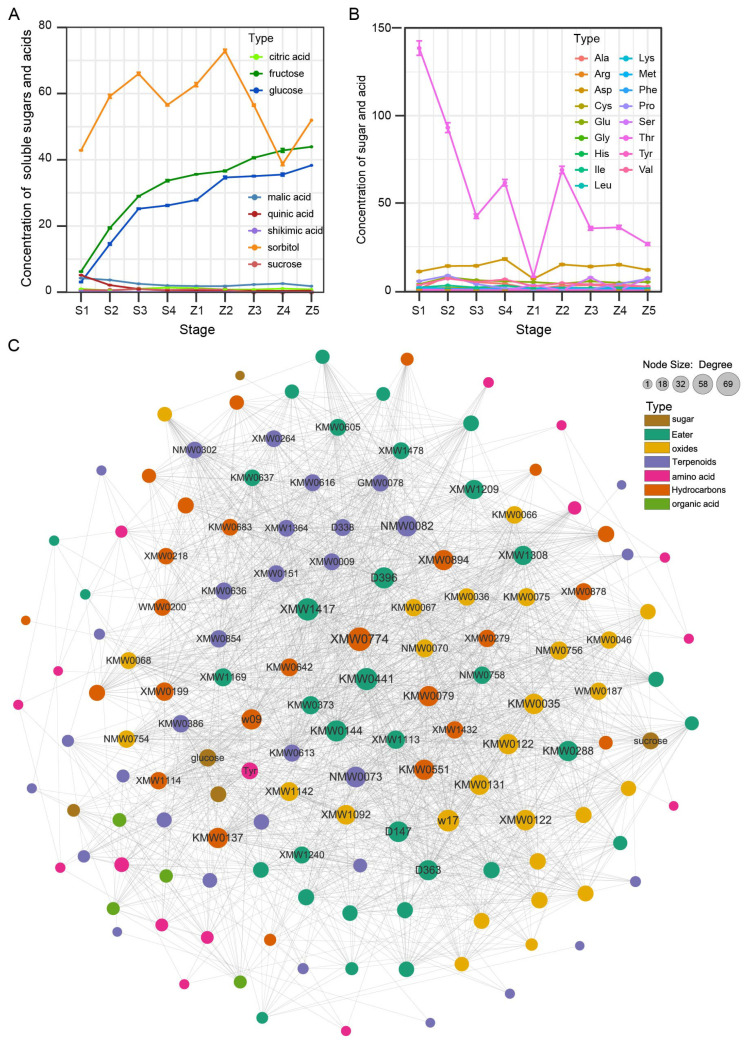
Three sugars and two amino acids are primarliy involved in aroma substances in pear fruits. (**A**) Concentrations of soluble sugars and acids from stage S1 to Z5. (**B**) Concentrations of amino acids from stage S1 to Z5. (**C**) Correlation network between soluble sugars or soluble acids and 119 key metabolites. The degree represents the neighbors per nodes.

**Figure 4 life-13-01504-f004:**
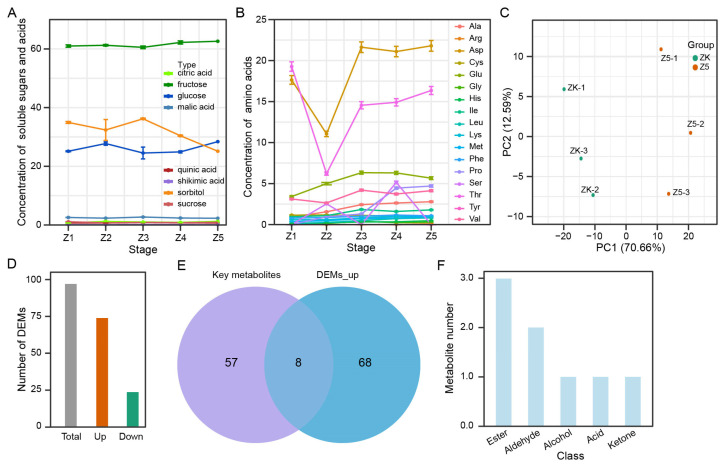
Eight metabolites screened out as dominant components in the ‘Panguxiang’ pear aroma. (**A**) Concentrations of soluble sugars and acids from stage Z1 to Z5. (**B**) Concentrations of amino acids from stage Z1 to Z5. (**C**) PCA of the ‘Panguxiang’ and the ‘Korla’ at stage Z5. ZP, the ‘Panguxiang’ at stage Z5; ZK, the ‘Korla’ at stage Z5. (**D**) DEMs of the ‘Panguxiang’ compared to the ‘Korla’ at stage Z5. (**E**) Veen diagram between 65 key metabolites and up-regulated DEMs. (**F**) Class of eight characteristic aroma substances of the ‘Panguxiang’ pear.

**Table 1 life-13-01504-t001:** Taxonomy of metabolites in the pear aroma metabolome.

Class	Number	Class	Number	Class	Number
Acid	7	Ester	97	Nitrogen compounds	4
Alcohol	34	Halogenated hydrocarbons	2	Phenol	5
Aldehyde	29	Heterocyclic compound	66	Sulfur compounds	3
Amine	4	Hydrocarbons	34	Terpenoids	91
Aromatics	19	Ketone	39		

**Table 2 life-13-01504-t002:** Eight characteristic aroma substances in the ‘Panguxiang’ pear.

Index	Formula	Compounds	Class I	Degree	cpd_ID	kegg_map
XMW1234	C5H8N2O3	(2E)-2-(Acetylhydrazono)propanoic acid	Acid	32	-	-
D456	C12H24O	8-Dodecen-1-ol, (Z)-	Alcohol	34	-	-
KMW0570	C12H22O	2-Dodecenal, (E)-	Aldehyde	34	-	-
KMW0475	C10H16O	(2E,4Z)-2,4-Decadienal	Aldehyde	30	-	-
KMW0373	C9H10O2	Benzoic acid, ethyl ester	Ester	43	-	-
NMW0203	C12H22O4	Diisopropyl adipate	Ester	34	C14523	-
KMW0580	C12H20O2	2,4-Decadienoic acid, ethyl ester, (E,Z)-	Ester	32	C08486	-
NMW0070	C9H8O2	1,2-Propanedione, 1-phenyl-	Ketone	42	C17268	ko00360, ko01100

## Data Availability

Not applicable.

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
