# Peer review of "Eight Typical Aroma Compounds of ‘Panguxiang’ Pear during Development and Storage Identified via Metabolomic Profiling"

_life, 2023, doi:10.3390/life13071504_

Round 1
Reviewer 1 Report
Dear Authors.
Your Article is interesting, very informative, has a scientific novelty and made a very positive impression.
Authors were investigated using HPLC the levels of sugars, organic acids, and amino acids in "Panguxiang" pear. Through metabolomics analysis at different stages of "Panguxiang" pear authors selected key methabolites that were significantly related to glucose, sucrose, fructose, tyrosine and Alyson according to the trends of metabolite concentrations.
But, despite the declared advantages, I would like to add a number of statements that may improve the article.
1. The final chapter Conclusions is missing. The Conclusions must be presented.
2. Authors write: Volatile aromatic substances are formed dynamically in fruits, and their types and concentrations change over time, but most of the aromatic substances are formed later stages.
In general, it would be good to hear a more detailed opinion of authors on this maxim, possibly constructed hypotheses.
3. Authors write: In the present study, we totally detected 434 metabolites from 14 types in "Panguxiang" pear, and majority of these metabolites consisted of esters and terpenoids.
Unfortunately, the article contains only a table with common groups of chemical compounds found in plant matrix. It would be very interesting and would significantly supplement the article if the authors deciphered, for example, the group of terpenoids.
4. Authors write: In addition, only Met content rose progressively from stage S1 to Z5, whereas four amino acids, including Try, Fly, The and Lys, were reduced gradually.
I also ask for a more detailed opinion of the authors, why such a process occurs.
And the end of my message, I would like to write, undoubtedly, authors did a good job on the stated topic and the article is worthy of publication in such a respected journal after minimal amendments.
Best regards.
Dear Authors.
The Article is interesting, very informative, has a scientific novelty and made a very positive impression.
Authors were investigated using HPLC the levels of sugars, organic acids, and amino acids in "Panguxiang" pear. Through metabolomics analysis at different stages of "Panguxiang" pear authors selected key methabolites that were significantly related to glucose, sucrose, fructose, tyrosine and Alyson according to the trends of metabolite concentrations.
Author Response
Response to the Reviewer #1
We are grateful to the reviewers for their constructive comments and valuable suggestions, which have helped improve the quality of the paper. All the comments have engaged and revised the manuscript carefully. The changes are visible in the revised manuscript. We used the “Track Changes” option to record the changes. We hope the reviewer will find the revised manuscript satisfactory.
Reviewer #1
Reviewer Comment 1:
Author response: Thanks. The last chapter Conclusions has added in Line 394-405.
Reviewer Comment 2:
Author response: Thanks a lot. We have added some characteristic metabolites in revised manuscript.
Reviewer Comment 3:
Author response: We have provided the list of all metabolites in the Supplementary Table 1.
Reviewer Comment 4:
Author response: Many thanks for the critical reading. Because the aroma of ‘Panguxiang’ pear was gradually appeared during storage stage, we selected the candidate metabolites with gradually change from developmental (S1) to storage (Z5) stage.
Thanking you
The Authors

Reviewer 2 Report
The authors of this manuscript present an interesting research study on Eight typical aroma compounds of ‘Panguxiang’ pear during development and storage identified by metabolomic profiling. The general purpose of this study was the ability to produce aroma volatiles that are important for 62 improving fruit aroma quality in this pear cultivar. Introduction and the rest of the sections are well described. I believe that the structure of the manuscript is in relation with the purpose of the manuscript. The presented tables and figures are clear; however, I think that some pictures needs to be enlarged. The authors discuss and conclude the findings of their work. The text needs few revisions. I believe that this review is very important as it describes the aroma compounds of ‘Panguxiang’ pear during development and storage. Thus, I believe that it can add further research interest.
Abstract
COMMENT:
According to my opinion the Abstract describe sufficiently the findings of this work.
Introduction
Introduction section is well written and, in my opinion, give the appropriate information without being extended.
Line 30 Pyrus
Line 40, 45 [3,4]. Please apply to the rest of the text [4,12,13]. (do not leave gap between number of references
Results
Figures 2 and 3 I believe that are too small
Discussion
No comments
Conclusions
Please add conclusion section
References
COMMENT:
Please check carefully the reference list according to the author’s instruction once again in order to be sure that is correctly typed and mentioned within the text.

Author Response
Response to the Reviewer #1
We are grateful to the reviewers for their constructive comments and valuable suggestions, which have helped improve the quality of the paper. All the comments have engaged and revised the manuscript carefully. The changes are visible in the revised manuscript. We used the “Track Changes” option to record the changes. We hope the reviewer will find the revised manuscript satisfactory.
Reviewer #2
Reviewer Comment 1:
Author response: Thanks for the reviewer’s suggestions. We had added more findings to the Abstract section.
Reviewer Comment 2:
Author response: Thanks for your valuable suggestions. We deleted this word in the revised manuscript. In addition, we have corrected the format of the references in the revised manuscript.
Reviewer Comment 3:
Author response: Thanks. We have revised the Figures 2 and 3 in the revised manuscript.
Reviewer Comment 4:
Author response: Thanks. The last chapter Conclusions has added in Line 394-405.
Thanking you
The Authors
